# Resting State Functional Magnetic Resonance Imaging Elucidates Neurotransmitter Deficiency in Autism Spectrum Disorder

**DOI:** 10.3390/jpm11100969

**Published:** 2021-09-28

**Authors:** Patrick J. McCarty, Andrew R. Pines, Bethany L. Sussman, Sarah N. Wyckoff, Amanda Jensen, Raymond Bunch, Varina L. Boerwinkle, Richard E. Frye

**Affiliations:** 1Division of Neurology, Section on Neurodevelopmental Disorders, Barrow Neurological Institute at Phoenix Children’s Hospital, 1919 E. Thomas Rd., Ambulatory Building, Phoenix, AZ 85016, USA; pmccarty@phoenixchildrens.com (P.J.M.); ajensen1@phoenixchildrens.com (A.J.); 2Mayo Clinic Alix School of Medicine, 13400 E Shea Blvd, Scottsdale, AZ 85259, USA; pines.andrew@mayo.edu; 3Division of Neurology, Barrow Neurological Institute at Phoenix Children’s Hospital, 1919 E. Thomas Rd., Ambulatory Building, Phoenix, AZ 85016, USA; bsussman@phoenixchildrens.com (B.L.S.); swyckoff@phoenixchildrens.com (S.N.W.); vboerwinkle@phoenixchildrens.com (V.L.B.); 4Division of Psychiatry, Barrow Neurological Institute at Phoenix Children’s Hospital, 1919 E. Thomas Rd., Ambulatory Building, Phoenix, AZ 85016, USA; rbunch@phoenixchildrens.com

**Keywords:** monoamine neurotransmitters, neurotransmitter deficiency, resting-state functional magnetic resonance imaging

## Abstract

Resting-state functional magnetic resonance imaging provides dynamic insight into the functional organization of the brains’ intrinsic activity at rest. The emergence of resting-state functional magnetic resonance imaging in both the clinical and research settings may be attributed to recent advancements in statistical techniques, non-invasiveness and enhanced spatiotemporal resolution compared to other neuroimaging modalities, and the capability to identify and characterize deep brain structures and networks. In this report we describe a 16-year-old female patient with autism spectrum disorder who underwent resting-state functional magnetic resonance imaging due to late regression. Imaging revealed deactivated networks in deep brain structures involved in monoamine synthesis. Monoamine neurotransmitter deficits were confirmed by cerebrospinal fluid analysis. This case suggests that resting-state functional magnetic resonance imaging may have clinical utility as a non-invasive biomarker of central nervous system neurochemical alterations by measuring the function of neurotransmitter-driven networks. Use of this technology can accelerate and increase the accuracy of selecting appropriate therapeutic agents for patients with neurological and neurodevelopmental disorders.

## 1. Introduction

Neurotransmitters are essential for normal brain development and function. Many neurological, neurodevelopmental, and psychiatric disorders have been shown to be associated with alterations in neurotransmitter function and neurotransmitter concentrations in the brain.

Autism spectrum disorder (ASD) is a very heterogenous neurodevelopmental disorder, primarily due to the fact that its diagnosis is based on behavioral observations which do not, at this time, have a precise neurological underpinning, although many brain systems have been implicated in driving ASD behavior. In fact, it has been proposed that ASD may have numerous underlying causes [1], leading to the optimal treatment of this disorder being difficult to determine without a cumbersome trial and error process [2]. Many different neurochemical alterations have been implicated in ASD. Cortical circuitry is believed to demonstrate an excitatory-inhibitory imbalance, implicating imbalances in the major cortical excitatory neurotransmitter glutamate and the major inhibitory neurotransmitter gamma-aminobutyric acid (GABA), as well as defects in GABAergic interneuron transmission and function [3]. Oxytocin, a key neurotransmitter believed to be involved in social motivation and bonding, has also been implicated in ASD and is undergoing intense study [4]. Finally, abnormalities in monoamine neurotransmitters including dopamine, norepinephrine and serotonin, which are sometimes caused by defects in biosynthesis due to deficiencies in the critical cofactors pyridoxal-5-phosphate [3], tetrahydrobiopterin [5] and/or folate [6], are associated with ASD.

Identifying abnormalities in monoamine neurotransmitters are particularly important as medication used to modulate dopamine, norepinephrine and serotonin are not only well developed but also actively studied in ASD [2]. However, as previously mentioned, ASD, like many neurodevelopmental and psychiatric disorders, is very heterogenous. Thus, much of the research does not always translate well into clinical practice because the research findings may represent a subgroup of patients, while clinical treatment aims to address the abnormality of the specific patient who may or may not be one identified in a specific subgroup. For example, although defects in the serotonin system have been implicated in ASD, simply treating the general ASD population with common medication to improve serotonergic neurotransmission does not appear to be effective and may even cause more harm than good in select individuals [7]. Thus, precision medicine has developed to assist in identifying biomarkers which can further guide a more personalized treatment approach.

The timely and accurate detection of neurotransmitter deficits can lead to impactful treatments for many neurological and neurodevelopmental disorders [8]. Current modalities with the capacity to detect alterations in neurotransmitters, such as positron emissions tomography (PET) and magnetic resonance spectroscopy (MRS), are limited in their spatial resolution and sensitivity to pertinent neurotransmitters, respectively [9]. Analysis of cerebrospinal fluid (CSF) neurotransmitter metabolites via lumbar puncture (LP) is the current standard screening method for suspected neurotransmitter deficits but is not without invasive procedural risks [10]. Hence, pharmacotherapies for suspected neurotransmitter deficits may be prescribed empirically in the absence of confirmatory testing. Therefore, a low-risk and clinically feasible screening biomarker of neurotransmitter deficits is needed to determine who may benefit from confirmatory LP.

The spatial distribution of neurotransmitter-associated brain networks is well established [11]. These networks and their cortical and subcortical spatiotemporal alterations can be detected by resting state functional magnetic resonance imaging (rs-fMRI) [11]. rs-fMRI has clinical applications in the evaluation of epilepsy [12,13,14] and other disorders [8,15]. Furthermore, rs-fMRI has been validated against other measures including intracranial electroencephalography (iEEG) [12] and task-based functional MRI in children [16]. Due to their widespread and targeted projections to spatially distant brain regions, these neurotransmitter systems can rapidly alter cortical network activity. For example, pharmacologic depletion of the dopaminergic system increases node-specific hemodynamic signal variability and decreases functional connectivity, suggesting that this system is important for the functional integration and stability of specific brain regions within large-scale networks. Thus, dysregulation of one or more of these neurotransmitter systems can produce broader network-level effects which can influence higher cognitive functions [17,18]. In the current case, rs-fMRI-detected network patterns revealed atypical deactivation of deep brain structures involved in monoamine synthesis, which was subsequently confirmed by analysis of CSF neurotransmitter metabolites.

## 2. Materials and Methods

rs-fMRI were acquired and analyzed as previously reported [12]. Acquisition was from a 3T MRI scanner (Ingenuity; Philips Medical Systems, Best, the Netherlands) with a 32-channel head coil at Phoenix Children’s Hospital. Acquisition consisted of two 10 min runs totaling 20 min. Parameters were 2000 millisecond repetition time (TR), 30 millisecond echo time (TE), 80 × 80 matrix size, 80-degree flip angle, 46 slices, 3.4 mm slice thickness with no gap, 3 × 3 mm in-plane resolution, interleaved acquisition, and 600 total volumes. For anatomical reference, a T1-weighted turbo field-echo whole-brain sequence was obtained with TR 9 milliseconds, TW 4 milliseconds, flip angle 80 degrees, slice thickness 0.9 mm, and in-plane resolution 0.9 × 0.9 mm. The Multivariate Exploratory Linear Optimized Decomposition into Independent Components (MELODIC) tool was used for characterization and visualization of individual resting state networks (RSNs), both typical and atypical, as previously reported [12].

## 3. Results

This case describes a 16-year-old female with a complicated medical history including developmental delay subtype of autism spectrum disorder (ASD), intellectual disability, severe hypotonia, developmental coordination disorder, mixed receptive-expressive language disorder, mitochondrial disease, immune dysfunction with a functional antibody deficiency, and irritable bowel syndrome with alternating diarrhea and constipation. She was born as a product of an in vitro fertilization triplet pregnancy requiring cerclage and terbutaline starting a 17-week gestation. She was born triple B at 34 weeks gestation by cesarean section. Her neonatal course was rather benign except for her prematurity, only requiring oxygen for 24 h and being discharged at day of life 17. She was born with a ventricular septal defect which closed spontaneously. Her triple sisters have a history of speech delay but have performed well academically. There is a history of depression and migraines on the maternal side of the family and thyroid disease and attention deficit hyperactivity disorder on the paternal side of the family.

At 2 years of age, she was diagnosed with global developmental delay and was diagnosed with ASD at 3 ½ years of age. She did not have any clear regression early in life but said several words in grade school but gradually lost expressive language, currently using word-like sounds and augmentative communication device to express herself. She can read some words and understand others and she is able to follow one and two step commands.

At about 12 years of age, anorexia developed with a loss of 25 lbs. In approximately 6 months self-injurious behavior started with knee banging and hitting hips to the point of bruising. Tic-like hand movements and ritualistic checking soon started. Workup for Pediatric Autoimmune Neuropsychiatric Disorder Associated with Streptococcus revelated elevated Streptococcus titers. Treatment with Augmentin, herbs and seven rounds of intravenous immunoglobulin improved the self-injurious behavior and tics-like movements and resulted in normalization of weight, but the severe repetitive behaviors continued. This deterioration was not associated with a clear clinical illness or seizures. She did manifest two brief staring episodes during this time, but an electroencephalograph was negative.

Her symptoms continued to progress. She progressively became more fatigued and stopped engaging in sports she enjoyed such as trampoline and swimming. She became more withdrawn with less family interactions and started to perseverate on YouTube videos. She developed chronic constipation requiring repeated cleanout. She developed period pain, including abdominal pain, headaches and leg cramps. Because of her continued deterioration she was referred to our center for further workup. Metabolic testing demonstrated laboratory values suspicious for mitochondrial disease and buccal enzymology (MITO-SWAB Religen, Plymouth Meeting, PA, USA), confirmed multiple complex deficiencies. Whole exome sequencing (Lineagen, Salt Lake City, UT, USA) demonstrated carrier status for a mutation in EIF3F (c.694T > G, p.Phe232Val) related to autosomal recessive EIF3F-related intellectual disability. She also underwent rs-fMRI due to late regression in cognitive abilities and worsening aberrant behavior.

Her rs-fMRI revealed aberrant and strongly deactivated networks localized to the brainstem and other subcortical structures involved in monoamine synthesis (Figure 1A). Additional aberrant networks extended over multiple bilateral regions including the ventromedial prefrontal cortex (vmPFC), the left fronto-temporo-parietal, and occipital networks with deactivation patterns localized to language and cognition-related regions (Figure 1B). Despite her deficits, she had typical frontoparietal (FP), language, and motor RSNs (Figure 1C). Suspicion for neurotransmitter deficiency was triggered given her history of prior severe late regression without known brain insult in the presence of aberrant deactivation patterns between the brain stem and vmPFC with a similar spatial distribution as the monoaminergic networks. Examples of typical rs-fMRI networks localized to the subcortical and brainstem networks (Figure 2A), vmPFC (Figure 2B), and language networks (Figure 2C) are provided for a case-control.

LP was performed to measure CSF neurotransmitter metabolites and rule out other neurometabolic disorders. Results (Table 1) were remarkable for decreased concentrations of 5-hydroxyindolacetic acid (58 nmol/L, reference range: 67–140 nmol/L) and homovanillic acid (117 nmol/L, reference range: 145–324 nmol/L), indicating decreased serotonin and dopamine synthesis, respectively. Pyridoxal 5′-phosphate (P5P), a required cofactor in the biosynthesis of serotonin and dopamine, was elevated at 82 nmol/L (reference range: 10–37 nmol/L). 3-O-methyldopa (18 nmol/L), 5-methyltetrahydrofolate (101 nmol/L), neopterin (12 nmol/L), tetrahydrobiopterin (16 nmol/L), and all amino acids were within reference ranges.

## 4. Discussion

We present a rare and unique case of neurotransmitter deficiency initially identified by objective evidence on rs-fMRI. Whole exome sequencing did not reveal a known inborn error of metabolism to account for this finding. Thus, the presumed underlying pathophysiology of her monoamine deficiency is acquired neuronal injury secondary to mitochondrial disease-mediated neurodegeneration. Workup of the origin of the mitochondrial disease is ongoing at this time.

Regional cerebral blood flow is temporally and spatially regulated by a process known as neurovascular coupling which involves the coordinated action of neurons and vascular cells to meet the high metabolic demand of the brain [19,20]. A decrease in monoamine neurotransmitters may cause reduced neurotransmission, resulting in deactivated network activity and a corresponding reduction in spatial oxygen utilization that can be observed on rs-fMRI as atypical blood-oxygen-level-dependent (BOLD) signals. Simultaneous acquisition of PET and rs-fMRI highlights the association between hemodynamic changes and neurotransmitter receptor density in relation to functional network organization, supporting the contribution of neurotransmitter systems to BOLD signal alterations [21,22].

While neurotransmitter systems and their association with large-scale brain networks are understood, and rs-fMRI has shown sensitivity to locations in the basilar brain regions [23,24], future work establishing the sensitivity and specificity of rs-fMRI biomarkers of neurotransmitter dysfunction is needed. Neurovascular coupling may differ between neurotransmitter systems due to intrinsic differences in mechanisms of neurotransmitter release, uptake, or transport, as well as release of small vasoactive molecules during these processes, like that reported between glutamate and dopamine [22]. Additionally, the dopaminergic modulation of prefrontal cortex NMDA receptor activity appears to be associated with the NMDA receptor co-agonist D-serine [25], suggesting a dynamic relationship between multiple neurotransmitter systems and other molecules that could further influence neurovascular coupling. The case herein likely has a relatively extreme clinical phenotype of severe regression, allowing for contextual interpretation of the atypical monoamine network patterns to inform the decision for invasive CSF confirmation with LP.

In addition to atypical subcortical and brainstem network patterns, this patient also had relatively deactivated language and cognition-related networks, commensurate with severe developmental regression. In children with ASD, there is evidence of atypical default mode network (DMN) connectivity profiles consisting of both hyper- and hypo-connectivity between specific nodes of the DMN as well as cross-network connectivity between the DMN and other large-scale brain networks, suggesting impaired functional intra- and inter-network integration in children with ASD [26]. Decreased or otherwise dysregulated neurotransmission secondary to neurotransmitter dysfunction may influence the existing network pathology resulting in such widespread network effect seen in the present case, as the monoaminergic deep brain structures are functionally integrated with broader networks such as the DMN [11]. Supportively, other monoamine disorders, such as chronic major depressive disorder (MDD), also demonstrate disease chronicity-associated BOLD signal reductions within the broader DMN [27], supporting the contribution of dysregulated neurotransmitter systems to large-scale networks over time. Thus, there may be a certain threshold of sustained neurotransmitter dysfunction that yields the relatively extreme network pattern we see in this case.

It is also possible that rs-fMRI may be useful as a biomarker of therapeutic effect, like that found in MDD with pharmacotherapy, wherein rs-fMRI showed an increase in functional connectivity within the DMN for patients with remitted MDD [27]. In patients with treatment-naïve MDD, rs-fMRI has been used to define biomarkers of depression subtypes and the differential response to various treatment modalities [28], which could greatly reduce the time and resources needed to identify the most appropriate and efficacious course of treatment. Additionally, characterization of DMN connectivity between specific nodes may correspond to clinical phenotype, wherein posterior cingulate cortex hyper-connectivity predicted social communication deficits in children with ASD [26], ultimately allowing for more targeted and personalized therapeutic interventions. Thus, rs-fMRI is being increasingly used in many different translational research initiatives, each with important clinical applications.

## 5. Conclusions

Numerous neurochemical alterations have been reported in individuals with ASD. Neurotransmitter systems play a crucial role in the overall functional relationship of large-scale networks and ostensibly impact the integration and modulation of these networks during development and beyond. rs-fMRI can be used to detect both typical and atypical RSNs, which can provide invaluable insight into the clinical implications of the neurochemical alterations associated with ASD. With continued research, rs-fMRI may soon emerge as a powerful tool in the diagnosis and management of various encephalopathies by characterizing biomarkers of neurotransmitter-driven network pathology, thus aiding in earlier detection of disease, and permitting the monitoring of response to treatment.

## 6. Patents

Nothing to report.

## Figures and Tables

**Figure 1 jpm-11-00969-f001:**
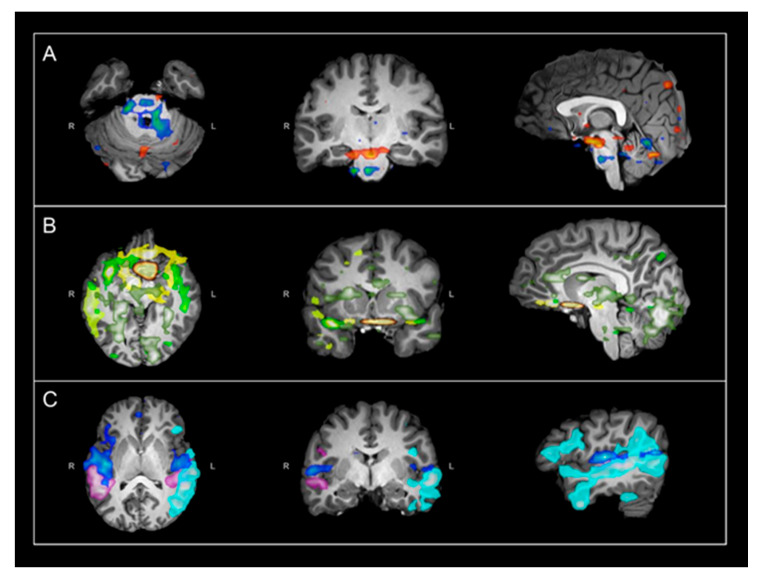
Select rs-fMRI networks in 16-year-old patient with ASD (case study). All images are in radiological orientation. The indicated networks are overlaid on T1-weighted images in axial, coronal, and sagittal views. (**A**) Evidence of atypical and strongly deactivated subcortical and brainstem networks in similar anatomic spatial distribution as the monoaminergic networks; (**B**) Additional atypical networks extending over multiple bilateral regions including vmPFC, FP, temporal, and occipital regions; (**C**) Normal language networks with bilateral presence of typical connectivity between receptive and expressive regions. Row A, blue color denotes BOLD deactivation, red color denotes BOLD activation. Rows B and C, each color denotes a separate network. Abbreviations: rs-fMRI = resting state functional magnetic imaging; vmPFC = ventromedial prefrontal cortex; FP = fronto-parietal.

**Figure 2 jpm-11-00969-f002:**
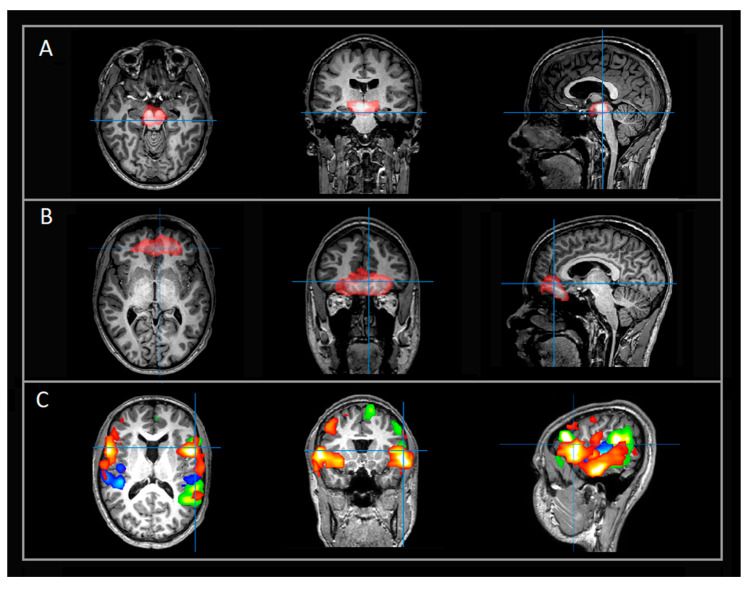
Select rs-fMRI networks in patient (case–control) evaluated for possible seizure focus (not detected, otherwise typical networks). All images are in radiological orientation. The indicated networks are overlaid on T1-weighted images in axial, coronal, and sagittal views. (**A**) Example of typical subcortical and brainstem networks; (**B**) Additional examples of typical networks extending over bilateral vmPFC region; (**C**) Typical language networks. Rows A and B, red color denotes BOLD activation. Row C, each color denotes a separate language network. Abbreviations: rs-fMRI = resting state functional magnetic imaging; vmPFC = ventromedial prefrontal cortex.

**Table 1 jpm-11-00969-t001:** Results of CSF metabolite analysis. * Abnormal results. WNL: Within normal limits.

Test	Result	Reference Rnge
5-Hydroxyindoleacetic acid	58 *	67–140 nmol/L
Homovanillic acid	117 *	145–324 nmol/L
3-O-Methyldopa	18	<100 nmol/L
5 –Methyltetrahydrofolate	101	40–120 nmol/L
Neopterin	12	8–28 nmol/L
Tetrahydrobiopterin	16	10–30 nmol/L
Pyridoxal 5 Phosphate	82 *	10–37 nmol/L
Amino Acids	WNL	-
Succinyladenosine	1.8	0.74–4.92 umol/L

## Data Availability

Data are available upon request.

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
