# Peer review of "Resting State Functional Magnetic Resonance Imaging Elucidates Neurotransmitter Deficiency in Autism Spectrum Disorder"

_jpm, 2021, doi:10.3390/jpm11100969_

Round 1

Reviewer 1 Report

Manuscript ID: jpm-1316221

Title: Resting State Functional Magnetic Resonance Imaging Elucidates 

Neurotransmitter Deficiency in Autism Spectrum Disorder

This manuscript demonstrated the usefulness of rs-fMRI for clarifying neurotransmitter deficiency in ASD. The authors have investigated their study with one specific patient, which I am quite concerned and I am not sure if the result they presented is reliable and reproducible.

More expanded rs-fMRI data analysis may also be required, e.g., by including the triple resting state networks (such as DMN, SN, CEN) and providing a comparison between the patient data and a group of healthy volunteer data would also strengthen the manuscript.     

Page 2 Line 78: why is the sensitivity of either PET or MRS poorer and limited? 

It should be Figure 1 (not Figure 2). Also, please include colour bar in the figure with an explanation. 

Author Response

1.This article is a case report, which is a report of a compelling association. Here we have demonstrated that the rsfMRI signal suggest abnormalities in deep brain structures responsible for neurotransmitter production is indeed correlated with abnormalities in neurotransmitter measured in the cerebrospinal fluid. Thus, the prediction of the rsfMRI findings is confirmed. This is a compelling finding that should spark more research in the future. As performing a lumbar puncture to measure neurotransmitters is not routine, the information presented is unique.

  1. We have now included a control patient for comparison. The resting state analysis provided reflects the clinical approach to identifying typical and atypical cognitive networks. More explained analysis of group data may be appropriate in expanded future studies.
  2. As outlined in the sentence, PET has limited spatial resolution and MRS has issue with the sensitivity to specific neurotransmitters
  3. The Figure label has been corrected and the caption now explains the color scheme in detail.

Reviewer 2 Report

Interesting and well written case report regarding the use of Resting State Functional Magnetic Resonance Imaging to Elucidate Neurotransmitter Deficiency in Autism Spectrum Disorder. 

Author Response

We thank the reviewer for the positive comments. 

Round 2

Reviewer 1 Report

I have no more concerns. Thank you. 

This manuscript is a resubmission of an earlier submission. The following is a list of the peer review reports and author responses from that submission.